# Specific Subtypes of Carcinoma-Associated Fibroblasts Are Correlated with Worse Survival in Resectable Pancreatic Ductal Adenocarcinoma

**DOI:** 10.3390/cancers15072049

**Published:** 2023-03-30

**Authors:** Karl Knipper, Alexander I. Damanakis, Yue Zhao, Christiane J. Bruns, Thomas Schmidt, Felix C. Popp, Alexander Quaas, Su Ir Lyu

**Affiliations:** 1Department of General, Visceral and Cancer Surgery, University Hospital of Cologne, Faculty of Medicine, University of Cologne, 50937 Cologne, Germany; 2Institute of Pathology, University Hospital of Cologne, Faculty of Medicine, University of Cologne, 50937 Cologne, Germany

**Keywords:** pancreatic ductal adenocarcinoma, subgroups of fibroblasts, personalized medicine

## Abstract

**Simple Summary:**

Despite massive research efforts, the mortality of pancreatic ductal adenocarcinoma is still high. In recent years, the tumor microenvironment has been revealed to play a key role in carcinogenesis. Therefore, our study aimed to further characterize the family of cancer-associated fibroblasts. We conducted stainings of four common fibroblast markers in a tissue microarray of 321 patients. Here, we describe three subgroups of expression patterns of these markers which are associated with worse survival. Following further basic research, this could lead to new targeted treatment options for patients with pancreatic ductal adenocarcinoma.

**Abstract:**

Purpose: The pancreatic ductal adenocarcinoma (PDAC) is one of the most lethal cancer entities. Effective therapy options are still lacking. The tumor microenvironment possibly bears further treatment possibilities. This study aimed to describe the expression patterns of four established carcinoma-associated fibroblast (CAFs) markers and their correlation in PDAC tissue samples. Methods: This project included 321 patients with PDAC who underwent surgery with a curative intent in one of the PANCALYZE study centers. Immunohistochemical stainings for FAP, PDGFR, periostin, and SMA were performed. The expression patterns of each marker were divided into low- and high-expressing CAFs and correlated with patients’ survival. Results: Tumors showing SMA^high^-, Periostin^high^SMA^high^-, or Periostin^high^SMA^low^PDGFR^low^FAP^high^-positive CAFs demonstrated significantly worse survival. Additionally, a high expression of SMA in PDAC tissue samples was shown to be an independent risk factor for worse survival. Conclusion: This project identified three subgroups of PDAC with different expression patterns of CAF markers which showed significantly worse survival. This could be the base for the further characterization of the fibroblast subgroups in PDAC and contribute to the development of new targeted therapy options against CAFs.

## 1. Background

Due to its prolonged life span, as well as its risk factor increase, pancreatic ductal adenocarcinoma (PDAC) is progressing in its importance as a tumor entity [1]. Since effective therapy regimes are lacking, PDAC continues to have a high mortality rate. In 2017, pancreatic cancer caused more deaths than breast cancer in the European Union [2]. The wide variety of risk factors supports the hypothesis that PDAC is not a homogenous tumor entity but should rather be considered as a heterogenous disease based on a great number of known pathomechanisms. A greater knowledge of biomarkers for PDAC in order to predict the patients’ survival or facilitate treatment decisions is needed.

In recent decades, the tumor microenvironment has received a large increase in research interest [3,4]. It has mainly been focused on the local and infiltrating fibroblasts, endothelial cells, and immune cells, such as macrophages, granulocytes, or lymphocytes [3,5]. Comparable to macrophages, fibroblasts build a cell type family consisting of a wide variety of subtypes with different functions—not only in the physiologic but also in the pathological cancer microenvironment [6,7]. Carcinoma-associated fibroblasts (CAFs) in PDAC derive from various cell types such as pancreatic stellate cells, tissue-resident fibroblasts, or tumor-infiltrating mesenchymal stem cells [8]. This cell family is the main source of extracellular matrix, which influences not only physiological but also pathological angiogenesis, nutrient supply, and cell migration [9]. Furthermore, CAFs play a fundamental role in the mechanisms of the immune escape of PDAC through the secretion of cytokines, chemokines, and growth factors [10].

Initially, CAFs in PDAC were considered solely as a barrier preventing cytotoxic chemotherapies from operating on the tumor cells. However, targeting the stromal myofibroblasts in human PDAC, for instance, resulted in an unexpected acceleration of disease progression [11]. It was shown in vitro and in vivo that differently regulated pathways and various cytokine expressions can be detected depending on the subtype of CAF in PDAC [12]. A duality in the fibroblast’s nature as a tumor-promoting and -suppressing unit, depending on the subtype, has arisen, making the further investigation of these cells important for improving modern personalized medicine [13]. The possibility of targeting a specific tumor-promoting subtype of fibroblasts could lead to an inhibition of therapy escape mechanisms while minimizing the occurrence of adverse events. Therefore, a deeper understanding of the fibroblast composition in PDAC is needed.

Several fibroblast markers are known. These markers are currently widely used in the clinical routine in different approaches depending on the specific enquiry.

α-SM actin (SMA) is involved in cell motility and identifies, among others, the myofibroblasts in normal, reactive, or neoplastic tissue. PDAC desmoplastic stroma normally demonstrates an abundance of myofibroblasts [14]. Here, SMA is expressed by activated myofibroblasts (so-called myCAFs), which are involved in the extracellular matrix reorganization and lead to an increased tissue contracture [15]. Evidence is rising that a higher tissue stiffness mediated by fibroblasts leads to higher regulated angiogenesis and therapy resistance [16]. These have an impact on PDAC patients’ outcomes, as there is a correlation between higher SMA expression and poorer survival rates [17].

Platelet-derived growth factor receptor-β (PDGFR) is involved in cell proliferation and migration. Stromal cells of mesenchymal origin, including fibroblasts recruited in pancreatic tumors expressing PDGFR, cause stimulation of tumor angiogenesis [18]. Nevertheless, PDGFR are also expressed in the normal endocrine pancreas and endocrine pancreatic tumors [19]. Despite their presence in both healthy and diseased tissue, PDGFR have been shown to be prognostic markers for worse survival, lymph invasion, and the occurrence of lymph node metastases in patients with PDAC [20].

Periostin is involved in the cell matrix organization and interacts with proteins of the extracellular matrix, therefore, resulting in remodeling and fibrosis [21]. Mostly known in the pathogenesis of cardiovascular and respiratory diseases, it has been shown to promote the invasiveness and resistance of pancreatic cancer cells [22]. Additionally, higher periostin expression in PDAC correlates with decreased patient survival and higher tumor stages [23].

Fibroblast activation protein (FAP) is known as a surface marker of CAFs and is overexpressed in different solid tumors, including PDAC, where its higher expression is associated with desmoplasia and a worse prognosis [24,25,26]. Immunohistochemical stainings on a tissue microarray of 134 patients with PDAC revealed a significant correlation between high FAP expression and worse survival rates [26]. Furthermore, analyses of tumor-surrounding fibroblasts showed a significant positive correlation between high FAP expression and the occurrence of lymph node metastases, disease recurrence, and death in patients with PDAC [27]. Targeting FAP-positive cells with a monoclonal antibody linked with a cytotoxic drug showed an effective inhibition of tumor growth in mice models of lung, pancreas, and head and neck cancer [28]. However, attempts to further target FAP-positive tumors in mice with T cells engineered with specific chimeric antigen receptors against FAP led to bone marrow toxicity and cachexia [29]. Hence, this underlines the fact that further characterization of fibroblast subgroups is needed.

The aim of this study is to describe this composition with the already established fibroblast markers FAP, PDGFR, periostin, and SMA in a big patient population in order to form further subgroups.

## 2. Methods

### 2.1. Patients and Tumor Samples

We included 321 patients who were operated on according to the German S3 guidelines for PDAC between 2013 and 2020 in one of the PANCALYZE study centers [30]. Formalin-fixed and paraffin-embedded samples were transferred by each study center to the University Hospital of Cologne. Tissue microarrays (TMAs) were assembled as described before [31]. Trained technicians created two 1.2 mm cylinders of each tumor sample with a semi-automated punch and transferred them to a paraffin block. For further staining, the tissue microarray was cut into 4 µm thick slides.

Written informed consent was obtained from each of the included patients. Data were collected prospectively according to the PANCALYZE study protocol and analyzed retrospectively [32]. Overall survival was defined as the time between the operative resection and the patients’ death or loss of follow up. Clinicians and pathologists of the study center captured the included clinicopathologic values following the 7th edition of the Union for International Cancer Control. The study was approved by the local ethic committees and was conducted in accordance with the Declaration of Helsinki.

### 2.2. Immunohistochemistry (IHC) and Analysis

Immunohistochemical stainings with antibodies against SMA, FAP, PDGFR, and periostin were performed following the manufacturer’s recommendations (Appendix A). Control stainings were conducted as mentioned in Appendix A. Stainings were performed automatically with the Leica BOND-MAX automated system (Leica Biosystems, Wetzlar, Germany). The obtained stainings were scanned with the Aperio GT 450 DX (Leica Biosystems, Wetzlar, Germany) and later analyzed digitally via QuPath v0.3.2 in a previously published manner [33]. The cell detection was performed under the following settings: setup parameters: detection image optical density sum, requested pixel size 1 µm; nucleus parameters: background radius 8 µm, median filter radius 0 µm, sigma 2 µm, minimum area 12 µm^2^, maximum area 400 µm^2^; intensity parameters: threshold 0.1, max background intensity 2; and cell parameters: cell expansion 5 µm, cell nucleus included. Both entire TMA samples were analyzed. The average value was calculated from these two samples for each patient. The total population was divided into a group with a high expression and a group with a low expression of each marker. The cutoff was defined as the median for FAP, PDGFR, and periostin. We defined the 45th percentile as the cutoff for SMA. Values lower or equal to the cutoff were defined as low.

### 2.3. Statistical Analysis

*p*-Values below 0.05 were considered statistically significant. All analyses were performed with IBM SPSS Statistics (version 28.0.1.1). Analyses of qualitative markers were performed with the chi-square test. Survival analyses were carried out with Kaplan–Meier curves. Interdependence of clinicopathologic values and survival was analyzed with the univariate and multivariate Cox regression analyses.

## 3. Results

A total of 321 patients with pancreatic ductal adenocarcinoma were included in this study. All patients were operated on with curative intention. Patient characteristics are shown in Table 1. The median follow-up period was 18 months. Of the cohort, 67.6% (*n* = 217) were over 64 years old. Seventeen patients (5.3%) received neoadjuvant (radio)chemotherapy. Tumor stage 3 was the dominant stage diagnosed in 53.3% of the study population. Of all included patients, 71% suffered from lymph node metastasis. Incomplete resection was diagnosed in 35.5%.

Fibroblasts are an important part of the tumor microenvironment. Therefore, we performed immunohistochemical stainings with the common fibroblast markers SMA, FAP, PDGFR, and periostin. Representative images of each marker are shown in Appendix A. Full tissue section validation in twenty tumor samples was performed to demonstrate the homogeneity of the marker expression. The four analyzed markers showed a homogenous expression.

We divided our patient cohort into low expression and high expression for each marker (SMA: *n* (low) = 145, *n* (high) = 176; FAP: *n* (low) = 161, *n* (high) = 160; PDGFR: *n* (low) = 161, *n* (high) = 160; periostin: *n* (low) = 161, *n* (high) = 160). After analyzing the stainings digitally, we looked for the interdependences between the different fibroblast markers and the overall survival. Here, we could show that a higher number of infiltrating SMA-positive cells correlates with a worse overall survival (*p* = 0.029, Figure 1). On the contrary, neither FAP nor PDGFR nor periostin stainings seemed to have an impact on survival when analyzed for the total population (FAP: *p* = 0.208; PDGFR: *p* = 0.237; periostin: *p* = 0.295).

We performed a Cox regression analysis to correct our results for effect modifiers. Here, overexpression of SMA was proven to be an independent risk factor for worse overall survival (HR: 1.389, CI: 1.019–1.893, *p* = 0.038, Appendix A and Table 2). Additionally, clinicopathological values such as T and N status are independent risk factors for worse overall survival (T status: *p* = 0.007, N status: *p* < 0.001, Table 2).

As we develop a deeper understanding of the tumor microenvironment, it is becoming clear that each cell type—such as macrophages or fibroblasts—is a heterogenous cell group playing various roles and expressing different markers. Therefore, we built further subgroups for different expression patterns of the four examined fibroblast markers.

Here, patients with Periostin^high^SMA^high^-expressing tumors were shown to have a significantly worse survival compared to those with Periostin^high^SMA^low^ tumors (*n* (low) = 56, *n* (high) = 103, *p* = 0.030, Figure 2A). Tumor samples with Periostin^high^FAP^low^PDGFR^low^SMA^high^ just showed a trend for a worse overall survival (*n* (low) = 15, *n* (high) = 16, *p* = 0.083, Figure 2B). The Periostin^high^SMA^low^PDGFR^low^FAP^high^ expression pattern was revealed to be an additional subgroup with a worse overall survival (*n* (low) = 15, *n* (high) = 9, *p* = 0.045, Figure 2C). All other combinations did not show an impact on the overall survival. No significant differences in patient characteristics between the groups Periostin^high^SMA^high^ and Periostin^high^SMA^low^ or Periostin^high^SMA^low^PDGFR^low^FAP^high^ and Periostin^high^SMA^low^PDGFR^low^FAP^low^ could be detected (Table 3). Representative consecutive pictures of each significant marker combination are depicted in Figure 3.

Similar analyses were conducted for the treatment-naïve subcohort. Here, we could observe similar results for the survival analyses as we showed for the whole patient cohort above (Figure 4).

Additionally, we could detect a significantly higher infiltration of PDGFR^high^ fibroblasts in neoadjuvant-treated patients (Table 4). In contrast, significantly fewer FAP^high^ expression patterns could be found in neoadjuvant-treated patients. No differences could be detected for SMA and periostin.

To compare the impact on patients’ survival of the three described subgroups of CAFs, we compared the median overall survival rate of the latter. Here, the subgroup with Periostin^high^SMA^high^ (we called them my-p-CAFs) showed the worst overall survival (SMA^high^: 20 ± 2.3 months, Periostin^high^SMA^high^: 15 ± 2.35 months, and Periostin^high^SMA^low^PDGFR^low^FAP^high^: 18 ± 4.85 months).

Recapitulated, this project described three subgroups of PDAC demonstrating worse patient survival—SMA^high^, Periostin^high^SMA^high^, and Periostin^high^SMA^low^PDGFR^low^FAP^high^. A high SMA expression was shown to be an independent risk factor for worse overall survival.

## 4. Discussion

This study investigated the role of four previously established fibroblast markers—FAP, PDGFR, periostin, and SMA—in a large study population of 321 patients with pancreatic ductal adenocarcinoma who underwent a surgical resection with a curative intension in one of the PANCALYZE study centers. The focus of this work was to characterize clinically relevant subgroups of CAFs by using a combination of these four listed markers. The importance of defining further subgroups of fibroblasts in patients with PDAC lies in the currently still limited treatment options. The use of FOLFIRINOX improved survival significantly [34]. However, the median disease-free survival is still 21.4 months [35]. Immunotherapy, which showed promising results in other malignancies, has not led to a breakthrough for patients with PDAC yet [36]. A depletion of FAP-expressing fibroblasts in in vivo experiments for pancreatic cancer showed synergisms with anti-PD-L1 therapy and led to tumor suppression [37].

We described that a higher infiltration of SMA-positive my-CAFs correlated with a worse overall survival in our study population. Additionally, a my-CAF-rich tumor stroma is an independent risk factor for a worse survival in patients with PDAC. This finding is supported by the literature [17]. Although many studies investigated different fibroblast markers in PDAC, only a few of them defined fibroblasts with a combination of these fibroblast markers. Öhlund et al. described two subgroups of fibroblasts. The FAP^+^SMA^high^ fibroblasts were located proximal to, and the IL6^+^SMA^low^ more distant from, the neoplastic cells [38]. In this study, we described Periostin^high^SMA^high^ CAFs (we called them my-p-CAFs), which showed a significantly worse survival in our study population. Additionally, we could define a subtype of CAFs, which is characterized by Periostin^high^SMA^low^PDGFR^low^FAP^high^, with a significantly worse overall survival compared to that of Periostin^high^SMA^low^PDGFR^low^FAP^low^. One should highlight that the sample sizes in these latter subgroups were small. However, thanks to the study design and the big study population, it was possible to form these subgroups. These findings are supported by several studies which characterized the impact on patients’ survival of fibroblast markers separately [17,23,26,39]. In a software-based manner, we could define as the first group, to the best of our knowledge, these CAF subgroups with a prognostic impact.

One of the limitations of this study was the inability to perform multiple stainings on a single tissue section, which could have provided more comprehensive insights into the underlying mechanisms, as the stainings described in this study were performed on consecutive sections. However, a direct overlay to describe the expression pattern of each cell was not technically possible. In future studies, immunofluorescent, sequential immunohistochemistry technics or multiplex immunohistochemistry could overcome this limitation [40]. Multiplex immunohistochemistry may offer a convenient method for evaluating the expression of multiple markers in individual cells while also providing spatial information. Once thoroughly implemented, this technique has the potential for use in clinical routine as it does not require any additional devices with high maintenance costs or extensive training of technician staff [41]. It was shown that multiplex immunohistochemistry is a more sensitive technique. However, a higher unspecific background staining could be detected, and a more thoughtful selection of antibodies must be kept in mind in order to avoid possible interactions [42].

One general limitation of the immunohistochemical analyses of the described markers was the lacking interpretation standardization of the staining. On the one hand, stainings are either analyzed qualitatively via optical microscope, semiquantitatively, or—as in this study—digitally via evaluation programs. On the other hand, no standardized thresholds are used in the literature for the four analyzed markers. After careful consideration, we decided to use cutoff values around the median. Given the absence of universally accepted cutoff values, we believe that this approach strikes a practical compromise. It is important to acknowledge that there is no consensus on this matter in the literature, as reported cutoff values for SMA in other studies range from 17% and 65% of the patient cohort [43,44,45]. This poses a challenge when comparing findings across studies. Therefore, it is crucial for the research community to address this issue and work towards achieving greater harmonization in this regard in the future. Additionally, using fresh tumor tissue or building organoids, followed by building single-cell suspensions with cell sorting via flow cytometry, could help to further investigate the role of each fibroblast subgroup regarding, among other things, angiogenic abilities and chemotherapy escape mechanisms. In breast cancer, four fibroblast subgroups could be identified by using several immunohistochemical stainings for similar markers as this study described [46]. Further investigations could show that these different subgroups not only occur in different kinds of breast cancer types but also show variations in immune cell infiltration and altered possibilities of the cell crosstalk [46].

Comparing the median overall survival of the three described CAF subgroups, we could show that my-p-CAFs (Periostin^high^SMA^high^) relate to the worst overall survival. In recent years, the role of fibroblasts in polarization-dependent tumor supporting or suppressing has been actively discussed [47]. Further characterization of the tumor-supporting fibroblasts subgroups could lead to new treatment options. A targeted inhibition of the tumor-supporting fibroblasts could possibly facilitate the propagation of the tumor-suppressing fibroblasts and, therefore, have an antitumor effect.

Taken all together, we were able to describe three subgroups of fibroblasts which showed a significantly worse survival. Especially, the fibroblast subtypes with Periostin^high^SMA^high^ and Periostin^high^SMA^low^PDGFR^low^FAP^high^ could bear the possibility of developing novel targeted therapy options for inhibition of these fibroblast subtypes without encountering severe side effects.

## 5. Conclusions

In this study, we were able to further describe the family of carcinoma-associated fibroblasts by using four widely established fibroblast markers. Three subgroups with specific expression patterns of these markers—SMA^high^, Periostin^high^SMA^high^, and Periostin^high^SMA^low^PDGFR^low^FAP^high^—demonstrated significantly worse overall survival. This, after further profound fundamental research, could lead to new personalized treatment targets, which could expand the current treatment options for patients with PDAC.

## Figures and Tables

**Figure 1 cancers-15-02049-f001:**
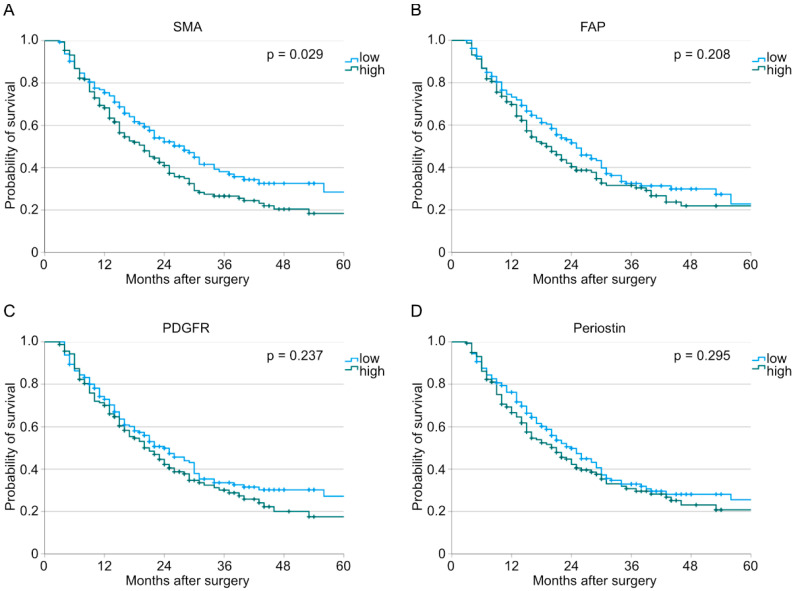
Analysis of overall survival performed with the Kaplan–Meier method depending on low or high expression of (**A**) SMA (*n* (low) = 145, *n* (high) = 176, *p* = 0.029), (**B**) FAP (*n* (low) = 161, *n* (high) = 160, *p* = 0.208), (**C**) PDGFR (*n* (low) = 161, *n* (high) = 160, *p* = 0.237), and (**D**) periostin (*n* (low) = 161, *n* (high) = 160, *p* = 0.295).

**Figure 2 cancers-15-02049-f002:**
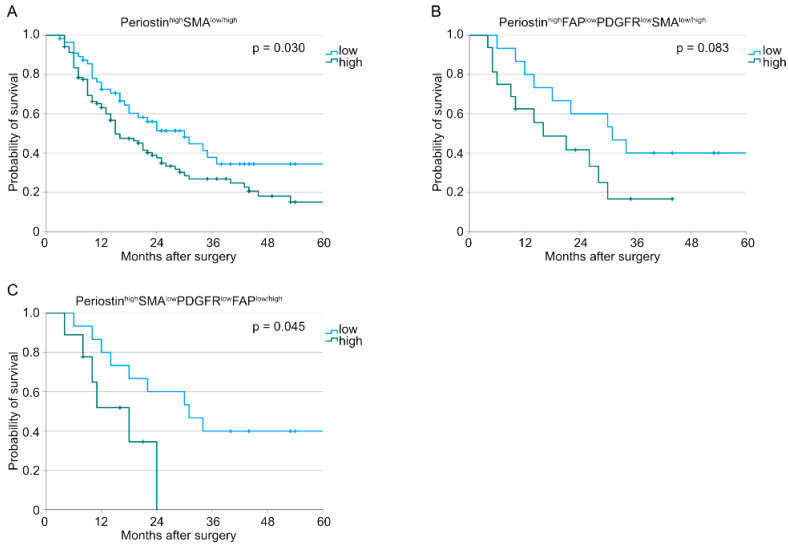
Kaplan–Meier curves for overall survival correlated with (**A**) Periostin^high^SMA^low/high^ tumors (*n* (low) = 56, *n* (high) = 103, *p* = 0.030), (**B**) Periostin^high^FAP^low^PDGFR^low^SMA^low/high^ tumors (*n* (low) = 15, *n* (high) = 16, *p* = 0.083), and (**C**) Periostin^high^SMA^low^PDGFR^low^FAP^low/high^ tumors (*n* (low) = 15, *n* (high) = 9, *p* = 0.045).

**Figure 3 cancers-15-02049-f003:**
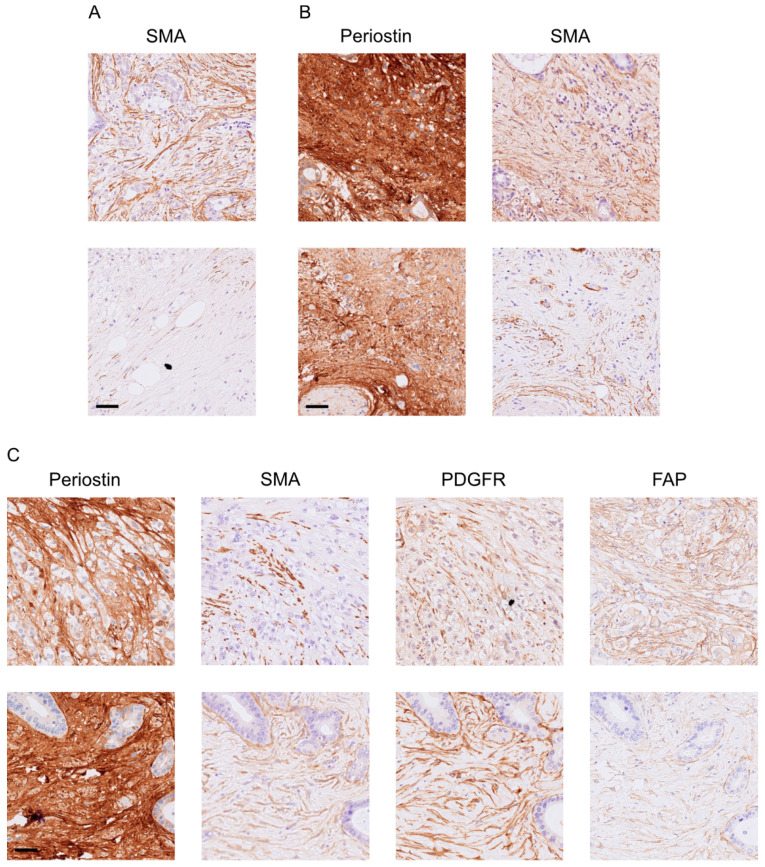
Representative consecutive pictures of patients (**A**) with SMA^high^ (**top**) or SMA^low^ (**bottom**) and (**B**) Periostin^high^SMA^high^ (**top**) or Periostin^high^SMA^low^ (**bottom**), as well as (**C**) Periostin^high^SMA^low^PDGFR^low^FAP^high^ (**top**) or Periostin^high^SMA^low^PDGFR^low^FAP^low^ (**bottom**). Sidebar: 50 µm.

**Figure 4 cancers-15-02049-f004:**
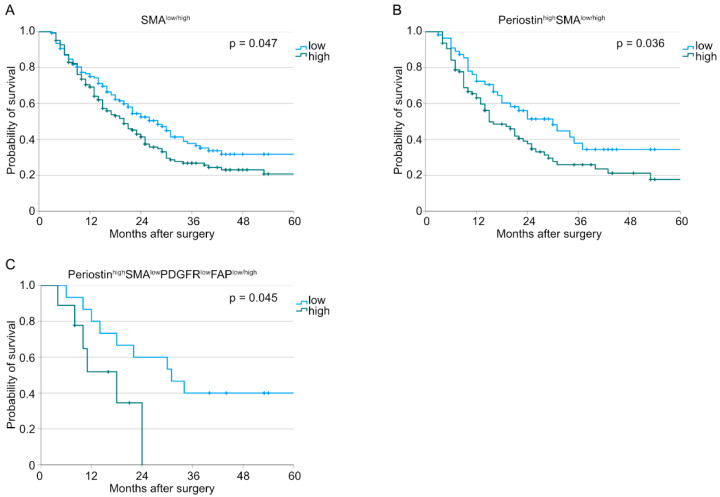
Kaplan–Meier curves for overall survival for treatment-naïve patients correlated with (**A**) SMA^low/high^ tumors (*n* (low) = 139, *n* (high) = 165, *p* = 0.047), (**B**) Periostin^high^SMA^low/high^ tumors (*n* (low) = 56, *n* (high) = 95, *p* = 0.036), and (**C**) Periostin^high^SMA^low^PDGFR^low^FAP^low/high^ tumors (*n* (low) = 15, *n* (high) = 9, *p* = 0.045).

**Table 1 cancers-15-02049-t001:** Characteristics of the total population of patients and SMA-low or -high subgroups. Bold print marks *p*-values below 0.05.

Characteristic	Total	SMA-Low	SMA-High	
	*n* (%)	*n* (%)	*n* (%)	
**No. of patients**	321 (100)	145 (100)	176 (100)	
**Sex**				0.270
Male	157 (48.9)	66 (45.5)	91 (51.7)	
Female	164 (51.1)	79 (54.5)	85 (48.3)	
**Age**				0.996
<65	104 (32.4)	47 (32.4)	57 (32.4)	
≥65	217 (67.6)	98 (67.6)	119 (67.6)	
**Median follow-up period (months)**	18	20	16	
**(range)**	(3–98)	(3–72)	(3–98)	
**Preoperative staging**				0.405
Primarily resectable	277 (86.3)	126 (86.9)	151 (85.8)	
Borderline resectable	38 (11.8)	14 (9.7)	24 (13.6)	
Locally advanced	1 (0.3)	0 (0.0)	1 (0.6)	
Unknown	5 (1.6)	5 (3.4)	0 (0.0)	
**Neoadjuvant therapy**				0.700
No	304 (94.7)	139 (95.9)	165 (93.8)	
Chemotherapy	14 (4.4)	5 (3.4)	9 (5.1)	
Radiochemotherapy	3 (0.9)	1 (0.7)	2 (1.1)	
**pT**				0.870
1	22 (6.9)	10 (6.9)	12 (6.8)	
2	121 (37.7)	52 (35.9)	69 (39.2)	
3	171 (53.3)	79 (54.5)	92 (52.3)	
4	7 (2.2)	4 (2.8)	3 (1.7)	
**pN**				0.324
0	93 (29.0)	46 (31.7)	47 (26.7)	
1	228 (71.0)	99 (68.3)	129 (73.3)	
**R**				0.262
0	207 (64.5)	88 (60.7)	119 (67.6)	
1	113 (35.2)	56 (38.6)	57 (32.4)	
2	1 (0.3)	1 (0.7)	0 (0.0)	
**Pn**				0.340
0	77 (24.0)	39 (26.9)	38 (21.6)	
1	232 (72.3)	103 (71.0)	129 (73.3)	
Unknown	12 (3.7)	3 (2.1)	9 (5.1)	
**L**				0.872
0	123 (38.3)	55 (37.9)	68 (38.6)	
1	195 (60.8)	89 (61.4)	106 (60.2)	
Unknown	3 (0.9)	1 (0.7)	2 (1.2)	
**V**				0.877
0	221 (68.9)	99 (68.3)	122 (69.3)	
1	94 (29.3)	43 (29.6)	51 (29.0)	
Unknown	6 (1.8)	3 (2.1)	3 (1.7)	

**Table 2 cancers-15-02049-t002:** Multivariate Cox regression. Bold print marks *p*-values below 0.05.

Characteristic	Borders	Hazard Ratio	95% Confidence Interval	*p*-Value
**Preoperative staging**				0.125
	borderline vs. primarily resectable	1.497	0.973–2.304	0.066
	locally advanced vs. primarily resectable	2.678	0.359–19.988	0.337
**pT**				**0.007**
	2 vs. 1	1.395	0.654–2.976	0.389
	3 vs. 1	2.316	1.091–4.916	**0.029**
	4 vs. 1	1.791	0.478–6.717	0.387
**pN**	1 vs. 0	2.177	1.449–3.271	**<0.001**
**R**	≥1 vs. 0	1.228	0.900–1.676	0.195
**Pn**	1 vs. 0	0.983	0.667–1.449	0.931
**L**	1 vs. 0	0.806	0.563–1.154	0.238
**V**	1 vs. 0	1.064	0.750–1.509	0.730
**SMA**	high vs. low	1.389	1.019–1.893	**0.038**
**FAP**	high vs. low	1.249	0.921–1.695	0.153
**PDGFR**	high vs. low	1.047	0.753–1.455	0.786
**Periostin**	high vs. low	1.078	0.790–1.471	0.637

**Table 3 cancers-15-02049-t003:** Patient characteristics for the Periostin^high^SMA^low/high^ and Periostin^high^FAP^low^PDGFR^low^SMA^low/high^ subgroups.

Characteristic	Periostin^high^SMA^low^	Periostin^high^SMA^high^		Periostin^high^SMA^low^	Periostin^high^SMA^low^	
				PDGFR^low^FAP^low^	PDGFR^low^FAP^high^	
	*n* (%)	*n* (%)		*n* (%)	*n* (%)	
**No. of patients**	56 (100.0)	103 (100.0)		15 (100.0)	9 (100.0)	
**Sex**			0.328			0.916
Male	27 (48.2)	58 (56.3)		7 (46.7)	4 (44.4)	
Female	29 (51.8)	45 (43.7)		8 (53.3)	5 (55.6)	
**Age**			0.411			0.562
<65	21 (37.5)	32 (31.1)		5 (33.3)	2 (22.2)	
≥65	35 (62.5)	71 (68.9)		10 (66.7)	7 (77.8)	
**Median follow-up period (months)**	21	14		31	11	
**(range)**	(3–65)	(4–73)		(6–65)	(4–24)	
**Preoperative staging**			0.735			0.235
Primarily resectable	48 (85.7)	89 (86.4)		12 (80.0)	9 (100.0)	
Borderline resectable	6 (10.7)	13 (12.6)		2 (13.3)	0 (0.0)	
Locally advanced	0 (0.0)	1 (1.0)		0 (0.0)	0 (0.0)	
Unknown	2 (3.6)	0 (0.0)		1 (6.7)	0 (0.0)	
**Neoadjuvant therapy**			0.101			-
No	56 (100.0)	95 (92.2)		15 (100.0)	9 (100.0)	
Chemotherapy	0 (0.0)	6 (5.8)		0 (0.0)	0 (0.0)	
Radiochemotherapy	0 (0.0)	2 (1.9)		0 (0.0)	0 (0.0)	
**pT**			0.463			0.742
1	7 (12.5)	6 (5.8)		1 (6.7)	1 (11.1)	
2	17 (30.4)	39 (37.9)		3 (20.0)	3 (33.3)	
3	31 (55.4)	56 (54.4)		10 (66.7)	5 (55.6)	
4	1 (1.8)	2 (1.9)		1 (6.7)	0 (0.0)	
**pN**			0.749			0.572
0	16 (28.6)	27 (26.2)		3 (20.0)	1 (11.1)	
1	40 (71.4)	76 (73.8)		12 (80.0)	8 (88.9)	
**R**			0.218			0.729
0	32 (57.1)	69 (67.0)		8 (53.3)	5 (55.6)	
1	23 (41.1)	34 (33.0)		6 (40.0)	4 (44.4)	
2	1 (1.8)	0 (0.0)		1 (6.7)	0 (0.0)	
**Pn**			0.841			0.526
0	12 (21.4)	23 (22.3)		3 (20.0)	3 (33.3)	
1	43 (76.8)	76 (73.8)		11 (73.3)	6 (66.7)	
Unknown	1 (1.8)	4 (3.9)		1 (6.7)	0 (0.0)	
**L**			0.783			0.562
0	24 (42.9)	41 (39.8)		5 (33.3)	2 (22.2)	
1	32 (57.1)	60 (58.3)		10 (66.7)	7 (77.8)	
Unknown	0 (0.0)	2 (1.9)		0 (0.0)	0 (0.0)	
**V**			0.487			0.285
0	35 (62.5)	68 (66.0)		10 (66.7)	4 (44.4)	
1	21 (37.5)	32 (31.1)		5 (33.3)	5 (55.6)	
Unknown	0 (0.0)	3 (2.9)		0 (0.0)	0 (0.0)	

**Table 4 cancers-15-02049-t004:** Comparison of the FAP, PDGFR, periostin, and SMA expression depending on neoadjuvant treatment. Bold print marks *p*-values below 0.05.

Staining	Treatment-Naïve	Neoadjuvant Treatment	
	*n* (%)	*n* (%)	
**No. of patients**	304 (100.0)	17 (100.0)	
**FAP**			**0.026**
Low	148 (48.7)	13 (76.5)	
High	156 (51.3)	4 (23.5)	
**PDGFR**			**0.024**
Low	157 (51.6)	4 (23.5)	
High	147 (48.4)	13 (76.5)	
**Periostin**			0.813
Low	152 (50.0)	9 (52.9)	
High	152 (50.0)	8 (47.1)	
**SMA**			0.400
Low	139 (45.7)	6 (35.3)	
High	165 (54.3)	11 (64.7)	

## Data Availability

The datasets generated and analyzed during the current study are available from the corresponding author upon reasonable request.

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
