# Peer review of "Specific Subtypes of Carcinoma-Associated Fibroblasts Are Correlated with Worse Survival in Resectable Pancreatic Ductal Adenocarcinoma"

_cancers, 2023, doi:10.3390/cancers15072049_

Round 1
Reviewer 1 Report
The article overall is well written and the topic of interest.
Minor criticisms :
- I would suggest broadening the background on carcinoma-associated fibroblasts in PDAC in order to support and to justify the choice of the four markers used for this study, possibly by providing more references.
- In the Discussion the limitations of the study were briefly mentioned, however the authors can elaborate more on the challenges and the opportunities provided by performing multiple stainings on the same section.
- Fig 3 and its legend could be improved as the staining images are so many and appear close to each other.
Author Response
The article overall is well written and the topic of interest.
We thank the reviewer for the very positive feedback.
Minor criticisms:
- I would suggest broadening the background on carcinoma-associated fibroblasts in PDAC in order to support and to justify the choice of the four markers used for this study, possibly by providing more references.
We have expanded our introduction to provide a more comprehensive overview of carcinoma-associated fibroblasts. Additionally, we have included more detailed information about the specific markers used in our study, with the aim of further clarifying our rationale for selecting them. We hope that these revisions adequately address the reviewers' concerns and improve the overall quality of our manuscript.
- In the Discussion the limitations of the study were briefly mentioned, however the authors can elaborate more on the challenges and the opportunities provided by performing multiple stainings on the same section.
In response to the reviewer's feedback, we have incorporated a new section in the discussion of our manuscript that specifically addresses the advantages and limitations of multiplex immunohistochemistry. We believe that this addition strengthens the scientific merit of our study and provides readers with a more well-rounded understanding of the potential applications and challenges of this technique.
- Fig 3 and its legend could be improved as the staining images are so many and appear close to each other.
We adjusted the design of Figure 3 according to the reviewer’s feedback.
Reviewer 2 Report
Line 249-250: Missing reference
I encourage the authors to validate their findings in atleast a subset of samples from independent cohort.
Other than SMA , are the other markers tested in this study specific to PDAC patients? The authors mention in their introduction (line 73-76) that Periostin is speculated to promote invasiveness and resistance to pancreatic cancer cells. In addition, the references mentioned for the role of FAP in PDAC are studies focused on cancer cell lines and not human subjects. I suggest the authors to do a thorough research about specific PDAC markers.
The authors mention – “The cutoff was defined as the median for FAP, 113 PDGFR, and Periostin. We defined the 40th percentile as the cutoff for SMA. Values lower 114 or equal to the cutoff were defined as low” . Why was a different threshold chosen for SMA?
The authors perform Cox-regression analysis on different features. Markers other than SMA are excluded from the analysis. The reason for exclusion is not mentioned. I encourage the authors to rerun the analysis with all the four markers in the study.
Survival curve for SMA alone is shown in the manuscript. It is hard to interpret the effect of other makers on survival . I suggest the authors to present survival curves for each of the other markers for interpretability.
Author Response
We thank the reviewer for the constructive comments, which we think improved our manuscript significantly.
Line 249-250: Missing reference
The missing reference was added:
Costa, A., et al. Fibroblast Heterogeneity and Immunosuppressive Environment in Human Breast Cancer. Cancer Cell 33, 463-479 e410 (2018).
I encourage the authors to validate their findings in at least a subset of samples from independent cohort.
We made efforts to contact our cooperation partners and other research groups in order to obtain access to an independent patient cohort. Unfortunately, we were unsuccessful in obtaining such access. However, we utilized a publicly available program for our analyses and detailed our settings in the methods section of our manuscript. This will allow other research groups to validate our findings on their own patient cohorts in future projects.
Other than SMA, are the other markers tested in this study specific to PDAC patients? The authors mention in their introduction (line 73-76) that Periostin is speculated to promote invasiveness and resistance to pancreatic cancer cells. In addition, the references mentioned for the role of FAP in PDAC are studies focused on cancer cell lines and not human subjects. I suggest the authors to do a thorough research about specific PDAC markers.
We conducted further literature research and adapted our introduction accordingly. The used markers are not only seen in PDAC and could be detected in other organs as well. Nonetheless, we have included multiple references for each marker to emphasize their relevance in PDAC.
The authors mention – “The cutoff was defined as the median for FAP, 113 PDGFR, and Periostin. We defined the 40th percentile as the cutoff for SMA. Values lower 114 or equal to the cutoff were defined as low”. Why was a different threshold chosen for SMA?
Unfortunately, there is currently no standardized method for analyzing SMA stainings. When reviewing the literature, we found a wide range of thresholds that are unstandardized and unexplained. Cutoffs used in previous studies range from qualitative classifications, such as low and strong stainings (resulting in exemplary subgroups of low vs. high: 17 % vs. 83 %, 13 % vs. 87 %, or 40 % vs. 60 %), to the use of semi-quantitative cutoffs such as < 10 % vs. > 10 % positive stromal cells (resulting in exemplary subgroups of low vs. high: 78 % vs. 22 %), or < 25% vs > 25 % positive stromal cells (resulting in exemplary subgroups of low vs. high: 65 % vs. 35 %, 28 % vs. 72 %). Other studies have used the mean (resulting in exemplary subgroups of low vs. high: 65 % vs. 35 %) or the median as a cutoff.
Sinn, M.; Denkert, C.; Striefler, J.K.; Pelzer, U.; Stieler, J.M.; Bahra, M.; Lohneis, P.; Dorken, B.; Oettle, H.; Riess, H.; et al. alpha-Smooth muscle actin expression and desmoplastic stromal reaction in pancreatic cancer: results from the CONKO-001 study. Br J Cancer 2014, 111, 1917-1923, doi:10.1038/bjc.2014.495.
Parikh, J.G.; Kulkarni, A.; Johns, C. alpha-smooth muscle actin-positive fibroblasts correlate with poor survival in hepatocellular carcinoma. Oncol Lett 2014, 7, 573-575, doi:10.3892/ol.2013.1720.
Chuaysri, C.; Thuwajit, P.; Paupairoj, A.; Chau-In, S.; Suthiphongchai, T.; Thuwajit, C. Alpha-smooth muscle actin-positive fibroblasts promote biliary cell proliferation and correlate with poor survival in cholangiocarcinoma. Oncol Rep 2009, 21, 957-969, doi:10.3892/or_00000309.
Kilvaer, T.K.; Khanehkenari, M.R.; Hellevik, T.; Al-Saad, S.; Paulsen, E.E.; Bremnes, R.M.; Busund, L.T.; Donnem, T.; Martinez, I.Z. Cancer Associated Fibroblasts in Stage I-IIIA NSCLC: Prognostic Impact and Their Correlations with Tumor Molecular Markers. PLoS One 2015, 10, e0134965, doi:10.1371/journal.pone.0134965.
Fujii, N.; Shomori, K.; Shiomi, T.; Nakabayashi, M.; Takeda, C.; Ryoke, K.; Ito, H. Cancer-associated fibroblasts and CD163-positive macrophages in oral squamous cell carcinoma: their clinicopathological and prognostic significance. J Oral Pathol Med 2012, 41, 444-451, doi:10.1111/j.1600-0714.2012.01127.x.
Luksic, I.; Suton, P.; Manojlovic, S.; Virag, M.; Petrovecki, M.; Macan, D. Significance of myofibroblast appearance in squamous cell carcinoma of the oral cavity on the occurrence of occult regional metastases, distant metastases, and survival. Int J Oral Maxillofac Surg 2015, 44, 1075-1080, doi:10.1016/j.ijom.2015.05.009.
Choi, S.Y.; Sung, R.; Lee, S.J.; Lee, T.G.; Kim, N.; Yoon, S.M.; Lee, E.J.; Chae, H.B.; Youn, S.J.; Park, S.M. Podoplanin, alpha-smooth muscle actin or S100A4 expressing cancer-associated fibroblasts are associated with different prognosis in colorectal cancers. J Korean Med Sci 2013, 28, 1293-1301, doi:10.3346/jkms.2013.28.9.1293.
Tsujino, T.; Seshimo, I.; Yamamoto, H.; Ngan, C.Y.; Ezumi, K.; Takemasa, I.; Ikeda, M.; Sekimoto, M.; Matsuura, N.; Monden, M. Stromal myofibroblasts predict disease recurrence for colorectal cancer. Clin Cancer Res 2007, 13, 2082-2090, doi:10.1158/1078-0432.CCR-06-2191.
Liu, L.; Liu, L.; Yao, H.H.; Zhu, Z.Q.; Ning, Z.L.; Huang, Q. Stromal Myofibroblasts Are Associated with Poor Prognosis in Solid Cancers: A Meta-Analysis of Published Studies. PLoS One 2016, 11, e0159947, doi:10.1371/journal.pone.0159947.
We defined our cutoffs based on the middle range of our digitally assessed cell numbers, which was resulting in a cutoff of the 45thpercentile for SMA.
We would like to express our gratitude to the reviewer for allowing us to address this important issue. As pointed out, further standardization of the thresholds for fibroblast markers is necessary. In response to this, we have included a further discussion section in our manuscript emphasizing the need for standardization.
The authors perform Cox-regression analysis on different features. Markers other than SMA are excluded from the analysis. The reason for exclusion is not mentioned. I encourage the authors to rerun the analysis with all the four markers in the study.
We revised the multivariate cox regression analysis with all four markers included. High SMA-expression proves to be an independent risk factor for worse overall survival (HR: 1.389, CI: 1.019 – 1.893, p = 0.038). The updated analysis was added to our results part.
Survival curve for SMA alone is shown in the manuscript. It is hard to interpret the effect of other makers on survival. I suggest the authors to present survival curves for each of the other markers for interpretability.
We added the survival curves of all four markers in Figure 1.
Round 2
Reviewer 2 Report
I appreciate the authors addressing the concerns